# Comparison of Metagenomics and Metatranscriptomics Tools: A Guide to Making the Right Choice

**DOI:** 10.3390/genes13122280

**Published:** 2022-12-03

**Authors:** Laura C. Terrón-Camero, Fernando Gordillo-González, Eduardo Salas-Espejo, Eduardo Andrés-León

**Affiliations:** 1Bioinformatics Unit, Institute of Parasitology and Biomedicine “López-Neyra”, CSIC (IPBLN-CSIC), 18016 Granada, Spain; 2Department of Biochemistry and Molecular Biology, Faculty of Sciences, University of Granada, 18071 Granada, Spain

**Keywords:** 16S, Bracken, Kraken2, metagenomics, metatranscriptomics, Nextflow, pipeline, QIIME2, shotgun sequencing

## Abstract

The study of microorganisms is a field of great interest due to their environmental (e.g., soil contamination) and biomedical (e.g., parasitic diseases, autism) importance. The advent of revolutionary next-generation sequencing techniques, and their application to the hypervariable regions of the 16S, 18S or 23S ribosomal subunits, have allowed the research of a large variety of organisms more in-depth, including bacteria, archaea, eukaryotes and fungi. Additionally, together with the development of analysis software, the creation of specific databases (e.g., SILVA or RDP) has boosted the enormous growth of these studies. As the cost of sequencing per sample has continuously decreased, new protocols have also emerged, such as shotgun sequencing, which allows the profiling of all taxonomic domains in a sample. The sequencing of hypervariable regions and shotgun sequencing are technologies that enable the taxonomic classification of microorganisms from the DNA present in microbial communities. However, they are not capable of measuring what is actively expressed. Conversely, we advocate that metatranscriptomics is a “new” technology that makes the identification of the mRNAs of a microbial community possible, quantifying gene expression levels and active biological pathways. Furthermore, it can be also used to characterise symbiotic interactions between the host and its microbiome. In this manuscript, we examine the three technologies above, and discuss the implementation of different software and databases, which greatly impact the obtaining of reliable results. Finally, we have developed two easy-to-use pipelines leveraging Nextflow technology. These aim to provide everything required for an average user to perform a metagenomic analysis of marker genes with QIMME2 and a metatranscriptomic study using Kraken2/Bracken.

## 1. Introduction

The human being is a complex assembly of approximately 40 trillion eukaryotic cells [1], which contain about 22,000 genes [2] and are tightly organised to form organs and tissues. In addition, the human microbiota, defined as a community of living microorganisms residing in a given ecological niche, is estimated to comprise as many as 100 trillion microbial cells [3], holding about 2 million genes [4]. Therefore, 99% of the genes that can be found in a human tissue pool are derived from microorganisms [5].

Other studies report that the number of human and bacterial cells are similar [6], and that both cell types maintain close contact [7,8]. A number of authors have also confirmed that the microbiome, defined as the set of microorganisms, their genes and metabolites in a given ecological niche, in adults is relatively stable and persists over time [9]. However, there are factors that can affect the microbiome, such as diet, probiotics and prebiotics intake, viruses, and drugs [9,10,11,12]. It has also been described that the microbiome suffers deep changes during pregnancy, particularly in the vagina and gut [13]. The gastrointestinal tract of the fetus is sterile, and microorganisms colonise the intestine in the course of delivery across the birth canal [14,15,16,17]. During childhood, the gut microbiome can be also influenced by several environmental factors, such as types of childbirth, geographic area, breastfeeding, and solid food regimen [18].

The gut microbiota is the most studied up to now, as it is known to influence virtually all human cells. In the last 5 years, according to PubMed, more than 4300 articles focusing on the gut microbiota have been published. This remarkable number represents a high percentage of the global publications in the field. As a result, it is now clear that microorganisms residing in the human gut play an important role in the metabolic processes of the host and can therefore be a potential source of new therapeutic strategies [19,20,21].

The main objective of this article is to summarise the knowledge about the importance of the microbiome in human health, disease and treatments. Moreover, we also aimed to provide an overview of the techniques and available tools in the field, both from the methodological and from the bioinformatics point of view. Finally, we also included two pipelines to facilitate metagenomics and metatranscriptomics data analysis, and we evaluated their performance using both simulated and experimental datasets. They are available at the GitHub repository: https://github.com/BioinfoIPBLN/16S-Metatranscriptomic-Analysis (accessed on 17 October 2022).

### 1.1. Microbiota: Human Health, Disease and Treatment

In recent decades, the microbiome has been studied to determine the involvement of these microorganisms in the development and prognosis of different diseases. Numerous studies and recent reviews have discussed different aspects of the microbiome and its possible role in human health, including in neonates and early life [22,23,24], but also regarding specific diseases such as cardiometabolic disorders [25], inflammatory bowel diseases [26,27], autoimmune diseases [28,29], neuropsychiatric diseases [30,31] and cancer [32,33]. In fact, it is now known that some gut bacteria can interact with human cells and, in particular, regulate the immune system [34] and mucosal immunity or inflammation [35] (see Figure 1). A wide range of metagenomic studies have focused on pathologies and their relationship with the treatment. Although this is a very mature field, there is much room for improvement, given that countless tissues and organs have not been studied in detail yet. The identification of the microbiota of each tissue and its association with diseases represents only one of the strategies implemented. Additionally, more knowledge about microbial composition diversity, microenvironments favoured by microbiota, or microbiota–host interactions, could contribute to the efforts against many diseases (see Figure 1).

### 1.2. Techniques Based on Next-Generation Sequencing

The main tools used to carry out this type of analysis (see Figure 2) have been made possible thanks to the advent of high-throughput DNA sequencing. The most commonly used techniques to detect microorganisms are meta-taxonomy, which refers to the sequencing of marker genes, and metagenomics, which refers to the random sequencing of microbial DNA [65]. Bacteria and archaea can be identified on the basis of the 16S subunit of small ribosomal RNA (16S rRNA), a gene that is distinctive for prokaryotic cells [66]. The equivalent method for detecting fungi is based on the use of nuclear ribosomal internal transcribed spacer (ITS), 18S rRNA or 26S rRNA regions [67]. Specifically, the 16S rRNA subunit has highly conserved regions common to the majority of bacteria, as well as unique hypervariable regions (V-regions) for each bacterial species. This allows the sequencing (using universal primers) and taxonomic identification of the bacteria that are present in a community, without amplification of human DNA [66,68].

Despite the sequencing of the 16S rRNA gene amplicon representing the primary tool for characterising bacteria in tissues with low bacterial biomass, this approach is limited by the challenges associated with short-read-length sequencing, including GC bias, sequencing errors and limited accuracy in taxonomic profiles at the species level [69,70]. For example, it is unable to detect more than 50% of species at the phylum Radiation, which represents 15% of the entire bacterial domain [71,72]. The technique is even more limited when it comes to defining and differentiating bacterial species between commensal and pathogenic strains [73], especially in the case of horizontal gene transfer. In addition, the selection of the variable regions to be analysed for each experiment must be carefully considered. This selection is essential because differences in the resulting microbial composition have been reported depending on the selected primer, as well as on the database and the bioinformatic tools used in the taxonomic assignment [66]. To overcome some of these technical limitations, other approaches rely on sequencers with longer reads, such as those provided by Pacific Biosciences (PacBio) and Oxford Nanopore (ONT), which can provide sequences exceeding 10,000 bp in length. These are currently being used for metagenomic assembly in low diversity communities [74] and the technique is expected to be improved for more complex metagenomic studies. 

Alternatively, shotgun metagenomics sequencing allows the study of the genomes in a microbial community, in order to determine their composition and provide insights into the biodiversity and functions of their components [75] (see Figure 2). After DNA extraction, it is fragmented and sequenced, so the use of specific loci as sequencing targets is avoided. Protein coding sequences from the metagenomic reads are then selected and compared with protein coding sequences from a reference database to obtain a functional profile. This method could be used to provide a profile describing the predicted biological functions discovered in the sequenced metagenome [75,76]. In addition, this methodology is able to detect viruses and viroids. Despite its numerous benefits, shotgun metagenomics sequencing has some limitations during DNA preparation and post-analytical processing techniques. Overall, the in silico inference of the metagenome has greatly improved the understanding of the microbial population dynamics and the contributions of the microbiota in the host. However, this technique does not offer any information about the microbial gene expression patterns that occur in response to environmental stimuli. Metatranscriptomic sequencing, however, can additionally identify mRNAs present in a microbial community, quantifying gene expression levels and providing the information needed to perform a functional study [77,78,79] (see Figure 2). Several studies have demonstrated that functional redundancy exists among related bacterial taxa and, as such, it is an important component of host fitness, as it has been described that functions can be conserved despite perturbations disrupting the balance of bacterial populations [80]. Moreover, metatranscriptomics allows sequencing of both the microorganisms and the host, so functional relationships can be established [64] regarding the observed phenotype on the basis of measurements of microbial activity and/or population dynamics [81]. 

As a complement, other-omic techniques, such as metaproteomics and metabolomics approaches, can be used to generate profiles of proteins and metabolites present in a sample [82,83]. Both techniques can identify metabolites and proteins that may mediate interactions between microorganisms [84]. However, the integration of multiple “meta-omics” can generate complex insights and involve comprehensive data analysis, requiring skills that may be lacking in most research groups [85]. 

### 1.3. Bioinformatic Tools

A number of bioinformatic tools have been designed to identify the microorganisms present in a sample. The information generated using high-throughput sequencing is becoming increasingly large and this poses a growing challenge for computational methods, which must minimise processing and memory requirements in order to provide a fast response and avoid overloading computational resources. Therefore, sample pre-processing is an essential step for a proper data analysis workflow. After quality analysis and removal of adapters and host data (if necessary), two main approaches can be highlighted: (1) classification of reads and (2) assembly of sequences (see Table 1 for details). All methods included in this classification rely on public databases, so consequently the content of these repositories and their quality have a large impact on the results and the interpretability of the microbiome.

#### 1.3.1. Pre-Processing

Pre-processing is usually the first and essential step in next-generation sequencing (NGS) analysis. The reliability of the results will be directly dependent on the quality of the dataset. There are plenty of tools to perform this step, mainly to identify and remove low-quality sequences and contaminants. Some of them are summarised below. 

##### Trimming and Quality Filter

The FastQC software [86] is employed for analysing the quality of reads, sequence length distribution and GC content distribution of each sample. The MultiQC software provides the possibility to summarise in a single report all information obtained by FastQC and others, so more accurate decisions are possible [87]. According to the report, it may be recommended to trim or filter out low-quality reads, as well as to remove adapter sequences. The cutadapt software allows users to remove adapter sequences, low-quality reads, primers, poly-A tails and other types of undesired sequences [88]. Similarly, Trimmomatic is a very flexible and efficient pre-processing tool that can correctly handle paired-end data, and trim and remove adapters matching the technology used for the sequencing process [89].

Another useful software is FastQ Screen which compares sequencing libraries with databases of whole genomes of organisms to infer whether the library composition matches the expected one [90]. Finally, BBtools trims and filters the reads using k-mers and entropy information, and also allows coverage normalisation by reducing the sampling of the reads (i.e., digital normalisation) [91].

##### Host Removal

This step is used to remove unwanted reads that align with any selected reference sequence, for example, those derived from hosts and/or possible contaminating factors, which all depend on the particular goals of the study and the applied technology. There are pipelines to perform most of these steps automatically. Among these, we can highlight kneadData [92] and miARma-Seq [93,94]. KneadData is a tool created to perform quality studies of metagenomic and metatranscriptomic sequencing data. In these experiments, samples are usually taken from a host, so the ratio of host to bacterial reads is high. This tool makes an in silico separation of bacterial reads from these “contaminating” reads, regardless of whether these come from the host, from bacterial 16S sequences or from other sources. miARma-Seq is a tool created to perform next generation sequencing (NGS) studies, and it can automatise many of the steps within the pre-processing of sequences, including filtering out low-quality sequences, removing adapters and separating host sequences.

#### 1.3.2. Taxonomic Identification

After quality control, reads can be directly submitted to taxonomic identification [65] using two main procedures: taxonomic classification or de novo assembly. The taxonomic classification of each read is a form of binning, as it groups reads according to their taxon ID. Binning can also be done by using other properties, such as compositional and co-abundance profiles. However, these methods usually require assembling the reads into longer contigs, which provide better results for profiling [95,96]. On the other hand, reads can be used for de novo assembly to obtain longer sequences, called contigs or scaffolds. This process is usually done when the sample may contain microorganisms that have not been correctly identified or have not been included in the available databases. Therefore, the choice among the approaches above mainly depends on the goal of the study, the samples and the knowledge of the microorganisms represented. In this section, we first performed a brief description of the main de novo assembly tools used in metagenomics (for more information see the references [65,97,98]), and then we focused on taxonomic classification in more detail.

##### Assembly

In sequencing experiments, specifically metagenomics experiments, hundreds of millions of reads can be generated in a single sample. Depending on the number of reads and the complexity of the microbial species to be studied, some libraries can be sequenced with enough depth, making possible the attempt of assembling the original genome sequence. In this case, as the sample contains multiple genomes, they require adapted algorithms, especially due to the unequal sequencing depth among organisms. In addition, it is very common for different strains of the same species to appear in the sample without a clonality event. Even so, assembly and binning of a metagenomics sample often succeeds in merging many of the reads, resulting in contigs that are easier to align to a genome reference database [98].

There are a number of assemblers that are based on assembling short reads into longer contiguous sequences. Some of them are MetaVelvet [99], MetaVelvet-SL [100] and Ray Meta [101], which are single k-mer Bruijn-graph-based assemblers for metagenomic data. IDBA (Iterative De Bruijn Graph Assembler) [102] and IDBA_UD first implemented this approach going from small k’s to large k’s, replacing reads with preassembled contigs at each iteration. IDBA-UD is a version of the IDBA assembler modified to tolerate uneven depth of coverage [103]. SPAdes [104] and MetaSPAdes [105] were developed for the assembly of single-cell cells, metagenomes and plasmids. These software work well with isolates and metagenomes but can be computationally expensive for any larger dataset [104,106]. MEGAHIT, which can be a fast and robust solution for large and complex metagenomic samples, uses a range of k-mers to iteratively improve assembly [107]. Finally, Bowtie2 or BWA-mem can be employed for validation of assembled contigs [95,107,108], along with a number of other tools. MEGAHIT [109] computes a number of statistics about the assembly errors and mismatches. CheckM [110] and BUSCO [111] also estimate both completeness and contamination of recovered genomes by employing lineage-specific single-copy marker genes and single-copy orthologs, respectively.

Methods based on de novo assembly were essential a decade ago, as microbial genome databases were not large enough. However, in recent years, lower sequencing costs have caused the number of near-complete genomes to increase exponentially [112]. Although a significant number of microorganisms remain to be characterised, we now find thousands of unique sequenced genomes accumulated for the application of reference-based methods [113].

##### Taxonomic Classification

Taxonomic classification tools compare sequences against a reference database of microbial genomes to determine sample composition. Early metagenomics analyses employed BLAST [114] to compare each read to all sequences stored in GenBank [115]. However, reference databases and the size of sequencing datasets have exponentially grown, rendering this strategy obsolete due to the large computational requirements. This has led to the development of metagenomics classifiers that provide much faster results, although at the expense of sensitivity compared to BLAST.

Currently, we find tools that may provide different information: some return a mapping of each read, while others provide the overall composition of the sample. Different strategies have been implemented to obtain this output, including read alignment, k-mers mapping, use of whole genomes, alignment of marker genes only or DNA translation, and alignment with protein sequences [65,116].

##### Taxonomic Profiles Based on Marker Genes

Taxonomic profiles based on marker genes arise from the identification of clade-specific gene-sets, so the identification of one of these genes is evidence that a member of the clade is present. The reference databases are much smaller, and therefore the assignment is much faster. The aligners are also quite sensitive. Some of them are Bowtie2 [117], used by MetaPhlAn, and HMMER [118], used by Phylosift [119] and mOTU [120]. The GOTTCHA tool [121] generates a database with unique genomic signatures based on unique fragments of 24 base pairs, which are indexed by bwa-mem [122]. This strategy should be accurate in abundance estimation, although problems arise in incomplete genomes where it is impossible to know the copy number. Other interesting tools are Mash [123] and sourmash [124], which use MinHash signature overlap. These tools allow estimating the similarity of datasets, so fast analysis at low computational cost is possible, leveraging the entire GenBank database. Among the tools for marker gene analysis, we must highlight QIIME 2 [69], as this software, together with its previous version, add up to 29 thousand citations. It provides an analysis platform based on ribosomal gene databases: 16S for bacteria, 18S for eukaryotes, and ITS for fungi. It is an open-source plug-in system where most of the programs to filter, trim, denoise and classify are included as “external” plugins. Some of these plugins, such as DADA2 [125] and Deblur [126], are designed for sequence quality control from different sequencing platforms, as well as for taxonomy assignment [127] and for phylogenetic classification [128]. Another reason for its success is that by working only with marker genes, the number of sequences needed to perform a reliable classification is very low, so the costs for sequencing samples that will later be analysed with QIIME 2 can be reduced [69].

##### Taxonomic Profiling Based on Whole Genomes and Transcriptomes

As mentioned above, read assignment is an important first step in taxonomic analysis, as it provides the basis for species identification and quantification. Therefore, here we elaborate on the assignment of metagenomic reads and transcript quantification from shotgun and transcriptomic data, in order to achieve rapid and accurate quantification of metagenomic strains [113,129,130,131,132].

There are many parallelisms between the analysis of transcriptomic data and the identification of microorganisms. For example, ambiguous genomic reads that are difficult to resolve at the strain level are analogous to the assignment of isoforms in RNA-Seq. The statistical issues at the heart of “comparative metagenomics” [133,134,135] are also similar to the challenges in differential expression analysis. In fact, the only relevant difference between metagenomics and RNA-Seq is that the sizes of the reference genomes are much larger than the transcriptomes. Another concept derived from RNA-Seq is the pseudoalignment, which was developed to take advantage of the fact that most of the statistics for RNA-Seq quantification are assignments of sequences to transcripts, rather than their alignments. Transcriptomic samples can be analysed using pseudoalignment techniques in the metagenomics setting [113]. In the context of metagenomics, as in RNA-Seq, the application of the expectation maximisation (EM) algorithm to “equivalence classes” allows accurate statistical resolution of mapping ambiguities. Furthermore, when combined with the EM algorithm, reads can be mapped much more accurately and quickly, allowing accurate statistical resolution of mapping ambiguities [136].

There are a number of tools for the identification of taxonomic profiles based on the taxonomic classification and quantification of nucleotides (see Table 1). The MEGAN software [137] was one of the first reference-based read mapping programs. This program provided phylogenetic context to mapped reads by assigning reads to the lowest taxonomic level that they could be uniquely aligned to. One of the disadvantages of MEGAN was that its approach to assigning ambiguous mapping reads limited its application to the quantification of individual strains. This strain identification not achieved by MEGAN was solved by the programs GRAMMy [138] and GASiC [139], which were the first to statistically assign ambiguous mapping reads to individual strains. However, these approaches relied on read alignment, demanding very high computational costs due to the huge size of the databases. ConStrains is another tool that identifies conspecific strains and reconstructs their phylogeny in microbial communities. This software uses single-nucleotide polymorphism patterns in a set of universal genes to infer within-species structures that represent strains [140]. Finally, the tool Taxonomic Analysis by Elimination and Correction (TAEC) uses the similarity in the genomic sequence in addition to the result of an alignment tool [141].

In 2014, it was demonstrated that it is possible to greatly accelerate read assignment using a fast k-mer hash, to avoid the need for read alignment [142]. Kraken and its improved version, Kraken2, employ this strategy for achieving the fast identification of all reads in a genomic sample. This algorithm is based on exact k-mer matches, so it replaces the alignment step by a simple search in a database containing every k-mer of every genome with the species identifier. If a k-mer appears in two or more taxa, the lowest common ancestor is stored. Kraken2 also provides direct support for 16S rRNA classification with any of the three standard 16S rRNA databases: Greengenes, SILVA, and RDP. In this manuscript, we used this feature to compare Kraken2 to the current software for 16S rRNA classification made by QIIME2 [143]. In addition, there is another tool called Bracken (Bayesian Reestimation of Abundance after Classification with Kraken2), which uses the taxonomic assignments made by Kraken2, along with information about the genomes, to estimate abundance at the species and/or the genus level. Bracken is able to adjust the abundance of identified organisms more precisely based on a Bayesian probability algorithm [144]. CLARK is another tool similar to Kraken2, which discards k-mers above genus. Both tools provide very sensitive results (i.e., very low false negative rate), although it will depend on the k-mers selected. We also found extensions of Kraken2 and CLARK that employ spaced or adaptive seeds that encode patterns for which only a subset of bases must match perfectly [145,146,147]. However, they have limitations in assigning different taxonomic levels. The Kallisto software [114] infers strain abundances by employing an expectation maximisation (EM) algorithm [148]. k-SLAM [149] is also a novel k-mer-based approach that uses local sequence alignments and pseudo-assembly, which generates contigs that can lead to more specific assignments.

Another interesting approach employs fast searches in amino acid databases. This may be the most sensitive classification of reads because amino acid sequences are conserved at much greater evolutionary distances than DNA. These tools require a higher computational cost. The two main tools based on this approach are DIAMOND [150] and Kaiju [151], which compare six-frame translations of reads with protein databases.

To sum up, direct taxonomic classification is very useful for quantitative profiling and identification of organisms with close relatives in the database. Furthermore, through this type of single-copy gene identification, in addition to classifying organisms, genome integrity and sample contamination can be measured [95,152].

**Table 1 genes-13-02280-t001:** Bioinformatic tools for metagenome and metatranscriptome analysis.

Preprocessing Tools			
Step	Tools	Description	Ref
Quality report	- FastQC	Reads quality, seq length distribution and GC%	[86]
	- MultiQC	Summarise results	[87]
	- FastQ Screen	Match a library with libraries expectation DB	[90]
Trimming	- Cutadapt	Find and remove adapters, primers, poly-A tails and others	[88]
	- Bbtools	Trims and filters by k-mers and entropy and downsampling reads	[153]
	- Khmer	k-mer error trimming	[154]
	- Diginorm	Downsampling reads	[155]
	- Trimmomatic	Trimming tool for Illumina	[89]
Host removal	- kneadData	Host sequences removal	[92]
	- miARma	Quality, trimming and host sequences removal	[93]
**Assembly for taxonomic classification**			
Assembly	- MetaVelvet	Single k-mer Bruijn-graph-based assemblers	[100]
	- Ray meta	Single k-mer Bruijn-graph-based assemblers	[101]
	- IDBA-UD	Multiple k-mers with preassembled at each interaction	[103]
	- Meta SPAdes	Multiple k-mers better assemblies with different abundances	[105]
	- MEGAHIT	Iterative k-mer fast and co-assembly robust metagenomic tool	[107]
	- CheckM	Evaluate quality of assemblies and contamination	[152]
	- BUSCO	Assess genome assembly and gene set completeness based on single-copy orthologs	[156]
**Taxonomic classification**			
Based on marker genes	- MetaPhlAn	Marker-gene-based taxonomic profiler	[157]
	- mOTU	Taxonomic profiler based on a set of 40 prokaryotic marker genes	[120]
	- GOTTCHA	Reads against unique subsequences at multiple taxonomic ranks	[121]
	- Mash	MinHash-based taxonomic profiler enabling super-fast overlap estimations	[123]
	- Sourmash	Dast searches with sequence bloom trees for taxonomic profiling	[124]
	- QIIME2	Completely re-engineered microbiome platform based on QIIME	[69]
Based on whole genomes and transcriptomes	- ConStrains	Uses single-nucleotide polymorphism patterns	[140]
	- TAEC	Uses the similarity in the genomic sequence and alignment tool	[141]
	- MEGAN	Uses BLAST or DIAMOND to match sequences and assigns LCA of matches	[137]
	- GRAMMy	Genome Relative Abundance using Mixture Model theory	[138]
	- GASiC	Genome Relative Abundance in a non-negative LASSO approach	[139]
	- Kraken2	*k*-mer search of reads against a DB built from multiple genomes	[142]
	- Braken	*k*-mer search of reads against a DB built from completed genomes	[144]
	- CLARK	Pseudo-assembled using k-mers using DB of nonoverlapping	[158]
	- Kallisto	Pseudo-assembled using k-mers using DB of nonoverlapping	[148]
	- k-SLAM	k-mer-based technique validated using the Smith–Waterman algorithm	[149]
	- DIAMOND	Spaced seeds with a reduced amino acid using protein homology search	[150]
	- Kaiju	FM index using classifier against protein sequences with reduced amino acid	[151]

### 1.4. Pipelines for Metagenomics and Metatranscriptomics Analysis

Microbiota analysis is a very complex and demanding task that requires the researcher to make a series of fundamental decisions in order to obtain accurate and reproducible results. Once the right tools have been selected and the parameters are defined, performing all the steps involved in the analysis in an optimal way can be complex and tedious. To avoid errors, an interesting approach is to create pipelines. These allow all the steps to be performed automatically, while being adaptable to the dataset of interest and the objectives of the study. Moreover, more than one tool is usually needed in the whole process, and it will be very often necessary to adapt the input and output formats to the requirements of each of the programs. The pipelines will then avoid the frequent errors when executing each of the steps and programs manually, in addition to the possibility of standardising and controlling each of the steps performed in a simple and detailed way.

Among the published pipelines for metagenomics and metatranscriptomics studies, it is worth mentioning Mothur as a pipeline which focuses on gene marks analysis. Mothur is an open-source software package which offers a list of commands to pre-process raw data for OTU-based analysis such as max length determination and removal of certain lineages [159,160,161]. It also provides a screen, pre-cluster and OTU-based analysis for taxonomic classification (k-mers). The standard global sequence alignment is computed using the Needleman–Wunsch algorithm. Mothur performs a taxonomic classification, OTU clustering, diversity analysis and visualisation of taxonomic levels (Krona and Phinch). In addition, for metatranscriptomic analysis, SAMSA has demonstrated a good performance. SAMSA is a pipeline designed for aligning metatranscriptomic sequences, which greatly increases its speed compared to BLAST [162]. It uses well-known tools such as PEAR and Trimmomatic for pre-processing and read trimming, SortMeRNA for filtering of ribosomal RNAs, DIAMOND for annotation, and R language for statistics tests and plots (DESeq). It also has an accurate prediction of organisms at the species level (~95%), and can predict both organism and function, which provides a complete picture of microbial gene expression. 

We propose here two pipelines: one for metagenomic analysis, based on QIIME 2, and another for shotgun and metatranscriptomic analysis, based on Kraken2 and Bracken. Both are running under Nextflow [163], which allows parallel execution and for the software requirements to be tailored. The fundamental programs in both pipelines are, according to the literature, the best for these kinds of analyses if they are parameterised correctly, which is why our pipelines are adapted according to the available data (see Section 2.2.4.). Moreover, we provide the code to verify the results generated by both pipelines, which will also help us to compare both techniques with each other. The details of both pipelines are discussed in Section 2.2.

## 2. Materials and Methods

### 2.1. Description of Three Datasets Used

Metagenomic reads provide a realistic test of performance; therefore, we analysed a case study of endometrial cancer that provides 16S and RNA-Seq data. However, these data do not allow an assessment of classification accuracy. This is the reason why two simulated metagenome datasets were also used.

#### 2.1.1. 10/100/400 Species Next Generation Sequencing Datasets

Mende and collaborators [164] generated a simulation of metagenomic sequences in order to test the performance of existing data analysis tools and methods. These data were generated with the iMESS-illumina metagenomic simulator software, for which three different genomic communities were created. These datasets are very useful for testing our Kraken/Bracken Pipeline, in order to identify the degree of accuracy in taxonomic identification at species level. Moreover, they also allow accuracy measurement through simulated genomes of microbial communities, which increase their complexity (10, 100 and 400 genomes). Three datasets are available in the following repository (http://www.bork.embl.de/~mende/simulated_data/, accessed on 17 October 2022). The three communities have an average of 26.66 million paired-reads and 75 bp of length. Further information regarding the dataset generation is described in the “Methods” section of Mende et al. [164]. Reference species reassigned to another taxon have been modified so that they can be compared with the database generated for taxonomic identification in the Kraken/Bracken Pipeline. Quality control was also conducted for all synthetic samples, showing low quality reads for every sample. Even so, our Kraken/Bracken pipeline managed to identify species quite precisely.

#### 2.1.2. Gut Microbiota Test Datasets

An adequate dataset for our purpose was the synthetic microbiome samples from the human gut generated by Almeida and collaborators (available at http://ftp.ebi.ac.uk/pub/databases/metagenomics/taxon_benchmarking, accessed on 17 October 2022). Two datasets, A100 and A500, were generated from 100 and 500 species, respectively, among the 66 most abundant genera across publicly available metagenomes from the human gut. The original datasets were constituted by fastq files from different hypervariable regions of 16S rRNA (V4, V12, V34, V45). The paired files obtained after concatenation of the different hypervariable regions contained 784,000 × 250 nt paired-end reads. Further information about this dataset is described in the “Methods” section of Lu and Salzberg and Almeida and collaborators [143,165]. We also performed quality control to verify that no trim process (adapters/primers/poly-A tails removal) was needed in our subsequent analysis.

#### 2.1.3. Description of Experimental Samples

This study from Li C et al. was conducted at the Shanghai First Maternity and Infant Hospital affiliated with Tongji University, from March 2018 to July 2020 [166]. A total of 40 women, including 30 endometrial cancer (EC) patients and 10 healthy controls (HCs), were enrolled. Endometrial tissue samples were obtained from these 40 women, who had undergone hysterectomy. Tissue samples from the 10 healthy patients, along with tumour samples from the 30 women, were studied using 16S. In addition, for these women with EC, adjacent non-tumour tissues were also obtained and studied in a paired RNA-Seq, together with the tumour tissue of the patients. All sequences can be obtained through the SRA portal: https://www.ncbi.nlm.nih.gov/bioproject/PRJNA750303 (accessed on 17 October 2022).

The RNA-Seq sequences provided by the authors were processed for adapter removal, so that the size ranges between 50 and 150 nt. The average size of the 60 paired-end libraries was 24,242,948. However, once the human sequences were removed with miARma-Seq, we obtained 1,027,228 of average size for the identification of possible microorganisms. In the case of the 40 16S samples, they displayed an average depth of 80,360 paired-reads and a length of 250 nt.

### 2.2. Software and Databases Used in the Analysis

#### 2.2.1. Kraken/Bracken/Krona

The tools used were Kraken2 (v2.2.1), Bracken (v2.7) and Krona (v2.7) (see Figure 3). For the analysis of the three datasets, Kraken and Bracken used two different databases, one from SILVA132 and the other from RefSeq. The database from SILVA was generated using the script named *16S_silva_installation.sh* and provided by Kraken2 for that purpose, which uses the small subunit NR99 sequence set. This program was slightly modified to download version 132, instead of the latest v138, since the taxonomic abundance values in the Almeida et al. dataset, which our results will be compared with, were obtained with version 132. According to the *kraken2-inspect* utility, 73% of this database contains minimisers that map to a taxon in the clade rooted at Bacteria, 22.30% map to the Eukaryota domain and the remaining 4.7% to Archaea. 

In the case of the RefSeq database, we used the *kraken-build* script to download the taxonomy and the datasets from Bacteria, Archaea, fungi, plant, protozoa and human. Moreover, to remove contaminant reads from sequencing projects, assemblies, linkers or primers, amongst, we also included vector databases from NCBI, such as UniVec and Univec_Core. The content of this database provided by the *kraken-inspect* tools is the following: 39.8% of this database contains minimisers that map to a taxon in the clade rooted at Bacteria, 58.7% to the Eukaryota domain, 1.13% to Archaea and 0.37% to viruses.

Once the databases were downloaded and indexed, Kraken2 assigned reads to the best location in a taxonomic tree contained in the reference database by invoking the following command: *kraken-db database Non-host.seqs.fq*. After this step, a report file was generated for subsequent analysis. 

To facilitate the visualisation of the results, the Krona-tools software is used to generate interactive graphs of the taxonomic classification from the Kraken2 output. These graphs cover all taxonomic levels of the classification, from kingdom to subspecies. To generate them, we used the *krona_import_taxonomy.py* script from the Krona-tools software. 

For abundance quantification, we used Bracken (Bayesian Reestimation of Abundance with Kraken), which derives probabilities describing how much of each sequence of a genome is identical to other genomes in the database. This tool is also capable of making assignments from a given sample to estimate abundance at the species and genus level, or even higher. This software also requires the building of a database adapted to the length of the reads, which is done by the *bracken-build* utility, by using the kraken database. In this work, three specific databases were built, adapted to 250, 75 and 50 nt, respectively. There are many parameters along this process, which can change the result of the analysis and can be modified in our pipeline (see Table 2).

#### 2.2.2. QIIME2

For the analysis of marker genes, we have incorporated QIIME2, as it is one of the most widely used software for this purpose. The QIIME2 part of our pipeline (see Figure 3) starts by importing the input data (fastq files) into a qza object, and demultiplexing if necessary. After this, DADA2 is used to denoise the samples, finally obtaining the representative features of the sequences. These features are mainly used for the taxonomic classification, generating the abundances of the microorganisms in each of the samples. For the conversion of these abundances to a readable content outside of the QIIME2 environment, biom functions are used.

Alternatively, other diversity analyses are also performed from the representative features of the sequences, such as α and β diversity, α rarefaction curve and taxa barplot, which are generated in qzv format (a specific file format for visualisation within the QIIME2 viewer).

There are many parameters along this process, which can change the result of the analysis and can be modified in our analysis pipeline (see Table 3).

#### 2.2.3. MetagenomeSeq

MetagenomeSeq (v1.36.0) is an R-package supporting statistical analysis that takes into account sparsity in OTU tables [167]. MetagenomeSeq has been developed to account for additional zero counts defining a zero-inflated Gaussian mixture model. The EM algorithm is used for the estimation of the fold change. Finally, significance assessment is performed by applying a moderated t-test, using the estimated fold change parameter in the mean model. This software has been selected because it outperforms other traditional methods [168]. This tool normalises the input data generated by the previous steps, corrects for the relevant covariates, and finally calculates the differential abundance between the experimental conditions.

#### 2.2.4. Nextflow

Nextflow is a reactive workflow framework and a programming domain-specific language (DSL) that eases the writing of data-intensive computational pipelines [164]. It provides a high-level parallel computational environment based on the dataflow programming model. An important advantage of this framework is its compatibility with diverse programming languages and different executing platforms, such as Simple Linux Utility for Resource Management (SLURM) [169], Amazon Web Service (AWS) [170], and docker container technology. These features make the developed pipelines reproducible, portable and easily parallelisable.

In this project, two pipelines based on Kraken2 and QIIME2 have been implemented using this framework (see Figure 3). Both pipelines generate a matrix with the microorganism abundance detected in each of the samples, which can be used for differential abundance analysis using MetagenomeSeq. Finally, to meet individual needs, our pipelines include different execution profiles with varying computational resources termed (1) “low”, which is intended to simulate an average laptop with 6 Central Processing Units (CPUs, 6 concurrent processes at maximum) and 24 GB of Random Access Memory (RAM), and (2) “high”, which represents a computational cluster with 36 CPUs and 156 GB of RAM. Execution times were compared in order to measure the improvement by the pipeline parallelisation capability, in terms of computational time (see Section 3.3). The command-line tool, documentation and Nextflow source code are available at the GitHub repository https://github.com/BioinfoIPBLN/16S-Metatranscriptomic-Analysis (accessed on 17 October 2022).

### 2.3. Dataset Processing

For each of the datasets described in Section 2.1, different analyses, pre-processing and filtering steps were carried out based on the characteristics of each study. In the case of the 10/100/400 species NGS data (Section 2.1.1), only a species-level abundance analysis was performed because abundance information was available for each species in each sample. For this reason, the only possible combination for achieving the detection of species in the different samples was the use of the Kraken2 pipeline and the RefSeq database. In addition, this study was not conducted with our QIIME2 Nextflow tool because the reads do not come from rRNA. However, the large number of reads in the samples meant that QIIME2 could not provide any result within a reasonable period of time. Due to the large number of paired-end reads (>26 million), it was also decided to set a confidence value of 0.5 for Kraken2, as the non-assignment of reads would not be a problem in the subsequent analysis. In addition, this parameter also ensures very reliable results for the most abundant microorganisms. This is of great interest in this study, since the objective is to compare against a reference abundance of a known number of organisms. These datasets had reads of 75 nt size, so a Bracken database adapted to this length was also used. Moreover, we had to review (NCBI taxonomic database; https://www.ncbi.nlm.nih.gov/Taxonomy/Browser/wwwtax.cgi (accessed on 17 October 2022)) the assignment of the apparently undetected microorganisms and check whether they had not been detected or had undergone a taxonomic reassignment. In the latter situation, taxonomic reassignments were manually performed on the reference data, in order to generate comparable results. The taxa with these modifications are included in Appendix A. 

Regarding the gut microbiota test dataset, the same reads were used to perform analyses with both the QIIME2 and the Kraken2 pipelines, using both the SILVA and RefSeq databases for the second. At the species level, only a comparison of the number of species detected by each software was performed, as we did not have absolute abundance information from the reference dataset at this level. As for the genus level, since we had reference abundance data [143], an analysis was carried out with both the QIIME2 and the Kraken2 pipelines, comparing their detection levels, as well as the correlation between the abundance results obtained and the real ones. In this case, the confidence value chosen for Kraken2 was 0 despite the probability of false positives, due to the low number of starting reads. Since this parameter reduces the number of successfully classified reads, a higher value would have caused a drastic impact on the results, which would hinder the objectives for this dataset. Since these data had reads of 250 nt size, a Bracken database adapted to this length was used. For the QIIME2 pipeline, it was necessary to select those lines that contained the taxonomic level analysed (D__5 for genus or D__6 for species), and to remove OTUs referring to uncultured or unidentified strains.

For the two analyses described above, the counts of the different OTUs belonging to the same species or genus were summed. Finally, the results were merged by selecting the OTUs present in the reference data from the output of each software. Concerning the OTUs not detected by our tools, the abundance value was set to 0. On the contrary, OTUs detected by our tools but not present in the reference data (false positives) were assigned to an additional category named “other”. Further information is available in Appendix A.

For the case study, with no reference to compare to, we processed 40 samples from 16S and 60 paired samples from RNA-Seq at the genus and species level. The QIIME2 pipeline was performed using the SILVA132 database. Once the abundance of each taxon per sample was obtained, we filtered to obtain the genus (D__5) and the species (D__6), removing OTUS in higher clades and uncultured/unidentified OTUS. In the case of the RNA-Seq samples, they were aligned by HISAT2 from the miARma-Seq pipeline, against the GRCh38.p13 human reference genome. By default, this tool generates paired-end fastq files containing the reads that do not align against this reference, forming the input of the Kraken/Bracken pipeline. In order to obtain the microorganisms in these samples, we used the RefSeq reference database, due to the fact that the Kraken/Bracken pipeline using the SILVA database did not yield any results at the species level. As previously mentioned, we used Kraken/Bracken to generate an abundance matrix of the microorganisms present in each of the 60 samples, for obtaining genus and then species. These data do not require any further post-processing. 

The three studies in this manuscript have been carried out using the default parameters of our pipelines, unless indicated otherwise.

### 2.4. Correlation Analysis

A correlation analysis for the two reference datasets was implemented. First, abundance values of the OTUs present in the reference data (real composition) were selected from the results by the different pipelines (if not detected, OTU abundance was set to 0). The Shapiro test was then applied to all datasets to assess the normality and select the most adequate correlation test. The Shapiro test showed that all datasets were non-normally distributed, so a Spearman test was selected to analyse the correlation between results and reference data. Correlation plots were also generated using the “ggscatter” function from the ggpubr R package [171].

## 3. Results

To standardise and optimise metagenomics and metatranscriptomics studies, we have created two pipelines that employ the up-to-date software that is considered to provide the best results. As an example, we used those pipelines in three different datasets, two of them considered to be the gold standard for validation of taxonomy classification. In these two cases, results were compared to the reference in order to determine the level of accuracy and their advantages/disadvantages. The third dataset was a case study using samples for endometrial cancer.

### 3.1. Simulated Samples

#### 3.1.1. 10/100/400 Species Next Generation Sequencing Datasets

The first dataset in this study contained 10 species with different abundances. After the reassignment of the reference taxa (see Section 2.3), we finally observed the detection of the 10 species with an abundance very similar to that expected (see Figure 4). Only one more species was detected by Kraken2 that was not present in the real data. However, this false positive OTU (*Methanococcus vannielii*) belongs to a genus that is present in the real data: Methanococcus. Furthermore, only 32 counts out of 16,818,916 (0.00019%) were assigned to this OTU, which is a negligible value compared to the other abundance values detected. We then reported a correlation coefficient of 0.97 between the abundance of the detected and expected species.

The second dataset from 100 species contained a number of strains, so after aggregating their abundance, the reference was reduced to 84 OTUs. The Kraken/Bracken pipeline, after filtering, detected 114 OTUs, 81 of which belonged to the reference data (three species were not detected; see Figure 5). Additionally, 33 false positive OTUs were detected, with 640,005 counts out of 17,632,395 total reads (3.63%, i.e., 19,394 on average per species). The correlation between the expected and estimated abundance of the detected species was 0.73. 

Finally, in the 400 species dataset, we expected to find 398 after aggregation. The Kraken/Bracken pipeline detected 464 species, 361 of which corresponded to the OTUs in the reference data, and 103 additional to false positive OTUs (see Figure 6). These false positives included 1,221,726 out of 17,643,183 counts (6.92%). In addition, 37 species contained in the reference data were not detected by the pipeline. The correlation between the expected and estimated abundance of the detected species in this case was 0.86 (see Figure 7). 

As a summary, for a study based on genomic sequences, as in this case of shotgun sequencing, the pipeline by Kraken2 leveraging the RefSeq database generates very good results, with correlation coefficients of 0.97–0.73. Out of the total number of sequences, only 3–6% are associated with species that are known to be not present in the samples, although they come from included genera. In addition, if deep sequencing is performed, there are parameters, such as --confident, that allow a clear separation between contained species and false positives. In these cases, it was only possible to use the Kraken2 pipeline because the datasets did not contain rRNA reads.

#### 3.1.2. Gut Microbiota Test Datasets

Almeida and collaborators [165] generated a dataset that has allowed several authors to test different metagenomics and metatranscriptomics tools, since these data are prepared to be analysed with a variety of software, including programs for 16S data analysis. Additionally, Lu and collaborators [144] provided data of high interest to identify the number of absolute reads associated with each of the identified genera for the 500 species dataset, and thanks to this, we can assess the quality of our analyses. Out of the 66 genera mentioned in the original paper, this last work only included the abundance of 58 of these genera, and they are the ones we will compare with the results of our pipeline.

After filtering and aggregating the results, the QIIME pipeline detected 88 genera. Alternatively, Kraken pipeline detected 175 genera using the SILVA database, whereas 193 genera were detected using the RefSeq database.

Out of the 58 genera in the reference (see Figure 8), QIIME was able to detect all, plus 30 additional genera, which represents 9.25% of the total counts (57,231 reads out of 618,860). Regarding the Kraken/Bracken pipeline using the SILVA132 database, 57 of the reference genera were detected, plus 84 additional genera with 57,234 out of 765,055 reads (7.48%). Regarding the Kraken/Bracken pipeline with the RefSeq database, we were able to detect 51 out of 58 reference genus, and 141 additional OTUs with 57,234 reads out of 780,133 (7.34%).

The correlation coefficient of the abundance of microorganisms between the sample and the reference was 0.67 using the QIIME 2 pipeline and the SILVA132 database, 0.82 with Kraken2 and SILVA132, and 0.65 with Kraken/Bracken when using the RefSeq database (see Figure 9; Appendix A).

Focusing on the species identified, we can highlight that the QIIME pipeline was able to detect 39 species in the A100 dataset and 213 in the A500 dataset (see Figure 10), which is far below the number of species contained in the reference datasets. Alternatively, the Kraken/Bracken pipeline using the RefSeq database was able to detect 143 and 419 species in the A100 and A500 datasets, respectively. Nevertheless, the Kraken/Bracken pipeline using SILVA132 did not manage to detect any species.

In this analysis, sequences from 16S hypervariable regions were used, so we can test both QIIME2 and Kraken2 pipelines. Based on the results, at the genus level the best option is to use Kraken2 with the SILVA database (see Figure 9) as it detects the majority of the genera with a low percentage of false positives. In this case, it is clear that using a whole genome database, such as RefSeq, against marker-gene sequencing data is not the best option. And after using SILVA as a reference database, the Kraken2 provides better results than those obtained by QIIME, since it identifies fewer false positives (7% vs. 9%) and the correlation of genera abundances are also better (0.82 vs. 0.67). It is worth noting than QIIME is quite good identifying the genera (58 from the 58) but not quantifying them (Figure 9). At the species level, where very similar organisms can be found, a large database containing the complete genomes is necessary to make a good classification. Therefore, Kraken’s results using RefSeq are much better than those of QIIME (see Figure 10).

### 3.2. Case Study—Microorganisms Detected in Endometrial Cancer Samples

The article by Li C. and collaborators [167] provided a suitable dataset for a comparative study among 16S and metatranscriptomic data samples from RNA-Seq.

#### 3.2.1. 16S Study

For this analysis, the 40 samples (30 from endometrial tumour tissue and 10 from endometrial tissue of healthy patients) were processed using our QIIME pipeline. We used different parameters to obtain both the genera (level 6) and the species (level 7). As for the genera, a total of 640 genus were obtained in the 30 tumour samples. In the samples from healthy patients, a total of 408 different genera were obtained (for more details see Appendix A).

Of these, 340 were shared between both types of samples. Due to their abundance, we can highlight microorganisms that are known to be typical of the endometrium, such as *Bacteroides*, *Prevotella*, *Pelomonas* or *Lactobacillus* [172]. We also obtained 300 genera that appeared to be present only in the cancer samples, as well as 68 from healthy patients. Among the most represented genera in the tumour samples, examples include *Anaerorhabdus, Spiroplasma* and *Gemmatirosa*. In addition, within the most abundant genera of the healthy samples we reported *Providencia*, *Lentibacter*, *Chelatococcus* and *Kineococcus*.

Beyond genus, at the level of species, 214 were obtained in the tumour samples, and 114 in the healthy tissue samples. Of these, 72 species were common (including *Streptococcus thermophilus*, *Rodentibacter pneumotropicus*, *Bacteroides plebeius* and *Lactobacillus iners* AB-1, and 142 were only present in cancer samples, such as *Nitrospirae bacterium*, bacterium NE3005, *Chitinophaga rupis* and *Catellicococcus marimammalium* M35/04/3. Finally, we reported 41 species abundant in the non-tumour tissue, including Lachnospiraceae *bacterium* 2_1_46FAA, *Pseudoxanthomonas helianthi* and *Prevotella intermedia*.

#### 3.2.2. RNASeq Samples

In the selected article, in addition to the 16S study, they carried out a paired transcriptome study of 30 women with endometrial cancer. To obtain those microorganisms present (at genus and species level), we removed all host sequences. Then, we used the Kraken/Bracken pipeline using the RefSeq reference database (see Methods), since SILVA132 did not provide species-level results.

In the case of the genus-level analysis, we found a total of 215 genera in the tumour samples, compared to 253 in the healthy tissue samples (see Appendix A for details). Of these, 188 are common genera, such as *Mycoplasma*, *Bacillus*, *Pseudomonas*, *Streptomyces* and *Acinetobacter* bacteria. In addition, viridiplantae genera (such as *Triticum*, *Gossypium* or *Chlamydomonas*) were very common. We also found 27 genera that only appear in the cancer samples, among the most abundant being *Rheinheimera. Rickettsia* and *Aquabacterium* were also present in lower abundance. Finally, among the 65 genera that appear only in healthy tissue, we found *Skermanella*, followed by *Janthinobacterium* and *Oxalobacter* (see Figure 11).

At the species level, we obtained a total of 224 in tumour tissue and 298 in healthy tissue. The 182 most abundant species shared between the samples included Triticum aestivum, Mycoplasma yeatsii and Bacillus cereus. Tumour tissue contained a total of 42 species, including Acinetobacter baumannii and A. schindleri, Pseudomonas alcaligenes and P. mendocina, or Ralstonia pickettii. Finally, there were 116 species present in the non-tumour tissue, and examples include Pseudomonas luteola, Massilia plicata or Escherichia coli (see Figure 12).

For the analysis of these data, we considered the most abundant genera and species from both 16S and transcriptomic data. In the case of 16S, we did not obtain results using Kraken2 with the SILVA database, mainly due to the small number of reads in the data (~80,500). Therefore, provided that the number of reads in a 16S study was very low, the best results would be obtained with QIIME2 rather than Kraken2, since it is able to identify genera and species with a very low number of sequences. With deep sequencing, as we show with the dataset by Almeida et al., it is advisable to use Kraken2 with SILVA. In the case of the RNA-Seq samples, the results above with the Mende et al. samples indicate that the use of Kraken2 with RefSeq is the best option (see Discussion for more details).

### 3.3. Computational Times

An important aspect to mention is the computational times required by each pipeline and study above. Overall, a better performance of the Kraken/Bracken pipeline could be observed, even when handling larger data, both in terms of sequencing reads and reference database.

In the case of the Almeida et al. data, all the pipelines proposed in this paper (Kraken/RefSeq, Kraken/SILVA and QIIME) were used to analyse the same dataset (A500 dataset) and therefore the run times are comparable with each other (see Figure 13a,b). We reported that the best results were obtained with the Kraken/SILVA pipeline, followed by Kraken/RefSeq and QIIME. The latter was approximately 7 times slower than Kraken/RefSeq and 70 times slower than Kraken/SILVA. However, the runtime improvement with parallelisation was not apparent in this case due to the low number of reads of this dataset.

The endometrial cancer dataset, however, contains a more reasonable number of reads, so this benchmarking is closer to reality. In this case, the computational times obtained by the Kraken2 pipelines, for both profiles, were lower than those by the QIIME pipeline (see Figure 13c,d). Within these, the use of the SILVA database v132 achieved much shorter times, mainly due to the large difference in size between the RefSeq and SILVA databases (120 Gb vs 500 Mb, respectively). Taking the QIIME pipeline run times with the high profile as a reference, Kraken/RefSeq was 5.4 times faster, while Kraken/SILVA was 344.5 times faster.

It is worth noting in this case that the total number of reads analysed by QIIME and Kraken were quite different (4,821,600 vs 62,199,120, respectively), so taking this variable into account, Kraken/RefSeq and Kraken/SILVA were 70.3 and 4444.2 times faster than QIIME, respectively.

More information about run times is available in Appendix A.

## 4. Discussion

When considering a metagenomics or metatranscriptomics project, the first thing to do is to set out the objectives to be achieved. Therefore, we must first consider the best sequencing technology. This choice is dependent on multiple conditions, but the available funding is likely to be a major determining factor. If the budget is low, performing metagenomics with 16S, 28S, ITS can be an interesting option. In this case, the selection of hypervariable regions (V-regions), which are unique for each bacteria, archaea or fungus, is very important. Depending on the sample to be studied, one region or another will be more appropriate, so it is strongly recommended to take this variable into account and make an informed selection of primers. Abellan-Schneyder et al. did an analysis for the gut flora, and their conclusions were that the most appropriate region was V3-V4, using the SILVA or RDP database. On the other hand, if the budget is higher, designing an experiment based on shotgun technology can be considered. This technology is capable of detecting many more microbial taxa, also including well-characterised viruses and viroids. However, there are still some limitations, because living microorganisms cannot be distinguished. If this is a relevant point in an experimental design, then metatranscriptomics may be preferable, as it overcomes the limitations of shotgun metagenomics sequencing [66]. On the one hand, it is able to perform microorganism identification, and on the other hand, it allows the study of spatiotemporal patterns of gene expression that occur in response to environmental stimuli. Therefore, metatranscriptomics would allow us to infer the functional or enzymatic capabilities of metabolically active microorganisms, as well as to obtain a picture of the relative abundance of the genes expressed in the host [173].

Once the technology and the appropriate experimental design have been selected, a set of bioinformatics tools that provide reliable and reproducible results must be chosen. After an exhaustive and deep literature search, we recommend the two software used in our two pipelines. For 16S analysis, we recommend using the QIIME 2 tool when a low number of reads exists, and Kraken2 when the library is large enough. It is worth noting that QIIME 2 is able to detect taxa with a low amount of biomass per sample, whereas other protocols such as shotgun or transcriptomic identify fewer taxa than expected [69]. 

When possible, an RNA-Seq providing transcriptome data will bring unique information by detecting expressed genes in the host, as well as biologically active microorganisms. In this case, we have based our pipeline in Kraken2 and Bracken. The choice of these programs is based on the precision efficiency provided by employing k-mer for read-level classification [143,174]. In that regard, the problem of ambiguous read-level classification is solved with Bracken, which generates more accurate estimates for species abundance in datasets already processed by Kraken2. Ye and co-workers did an extensive tool review of 20 metagenomic classifiers using simulated and experimental datasets. Their first conclusion was that DNA-based classifiers, such as Kraken2, provide better estimates than protein-based classifiers, such as Kaiju, DIAMOND and MMseqs2, when using a uniform database for these whole genome datasets.

Computationally, when a server with large amounts of RAM memory (>100 Gb) is available, Kraken2 provides good performance metrics and is very fast on large numbers of samples, as long as the database load time is amortised. It also allows the creation and use of custom databases. Moreover, our pipelines confirm these statements, as the pipeline based on Kraken/Bracken was 344 times faster than the one based on the QIIME2 platform, using the same database and achieving more accurate results (correlation coefficient from 0.82 versus 0.67). Moreover, QIIME2 requires a high amount of computing resources, so it is almost unachievable to analyse datasets with more than 50 million reads. CLARK is a good alternative, but the combination of Kraken2 and Bracken tends to have slightly more accurate abundance profiles. MetaPhlAn2 and Centrifuge may be also useful, despite the shortcomings of the default compressed database [175]. Tools such as MEGAN, GRAMMy or GASiC were discarded due to certain drawbacks such as poorer results in assigning low taxa. They also require high computational costs without a significant impact on the results. However, they do seem to work better with long reads, which is not the objective of this pipeline. The ConStrains tool was not applicable for our target due to its high specificity.

The biggest problem, no matter the technology and the pipeline selected, is the number of the false positives, a problem that is reflected in other works such as those of Ye and collaborators and Li [144,175]. The most recommended approach is to filter out false positives of low abundance, using a given abundance threshold, as we have seen work perfectly in the i10 dataset by Mende. Another approach could be to filter by reads according to the alignment score. However, it is complex to set these thresholds as they vary depending on the classifier and experimental design. To decrease the number of false positives, we recommend, in the pre-processing phase of the samples, to remove host-derived reads [176]. It is also recommended to include contaminant genomes in reference databases to reduce misclassification errors due to a lack of reference sequences. Clean laboratory practices, as well as recent innovations to experimentally remove routine contaminants [177], or the use of artificial sequences to quantify contamination [178], may reduce this problem, but it is unlikely that they can eliminate it completely. Another interesting option is the use of specific databases to decrease the number of false positives. In that regard, the rapid growth and taxonomic reassignments in reference databases pose a challenge for microbiome analysis, as there are frequent changes in the taxonomy database of NCBI such as renaming, deletion and merges between taxa of different levels. Therefore, when comparing our generated data with updated databases, some reads were assigned to unexpected taxa (false positives), which are not. All this leads us to the reflection made by many other authors, of the complexity that exists when trying to compare experiments performed with different bioinformatics tools and databases. Another possible explanation why these false positives may occur is the discrepancies in the concept of hierarchical taxonomy when applied to microorganisms, as it was initially formulated for organisms with sexual reproduction and no horizontal gene transfer. Therefore, microorganisms may violate well-established assumptions, which may result in sequences being assigned to erroneous identifiers [66]. For example, the same taxonomic level may contain different levels of sequence similarity [179,180]. The consequence of this variability for computational classifiers is that, at the species or genus level, different levels of sequence similarity in different parts of the taxonomic tree have different meaning, making it impossible, for some taxa, to design consistent rules that assign reads or contigs to a species, and there is clearly no fixed percentage identity threshold that can be used to group sequences into the same species or genus. Furthermore, in fungal taxonomy there is a multiplicity of names for the same organism and, consequently, metagenomic classifiers could assign the sequences to either taxon, and both would be correct, even though they appear to be different species [181,182]. Another issue is taxonomic changes, as there are no versions and this makes comparison of analyses over time complex [164]. In addition to the above, viruses have their own specific problems to be studied. They do not have universally conserved genes, in addition to presenting a greater diversity than in bacteria [183] and a high mutation rate. Faced with this rapid growth in the variety of viral species, a scientific consortium proposed a new framework for incorporating viruses discovered by metagenomic sequencing into the official taxonomy of the International Committee on Taxonomy of Viruses [184]. It is recommended to adapt alignment algorithms to allow the identification of more mismatches [185]. 

Considering all of the above, we can clearly see the importance of database selection and its relation to the appearance of false positives. This is helped by the fact that unculturable bacteria are classified by receiving the name “candidatus” followed by the putative name and species, or are named only informally without being covered by the standard nomenclature [151,164,183]. The NCBI taxon “Unclassified Bacteria”, which contains several candidate divisions, is placed directly under the “Bacteria” taxon node, thus falling outside the taxonomic hierarchy. In addition, GenBank and the BLAST nr/nt database [186] contain thousands of “unclassified” sequences, especially from metagenomes assigned to a taxonomic ID. The shared sequences of such correctly placed taxa and organisms can pose a challenge to metagenomic methods that attempt to cluster sequences or calculate the lowest common ancestor. Especially when using BLAST nr/nt or nr databases, it may be useful to filter out unclassified sequences, or to include only microbial taxa, as the kaiju classifier [151] does by including non-redundant eukaryotes.

In this work we have developed two pipelines based on Nextflow technology, which incorporate widely used software. Subsequently, we have applied them to three different datasets, to be a guide for inexperienced users. In the first dataset, which can only be analysed by Kraken/Bracken because it does not contain rRNA sequences, we observed that this pipeline is quite accurate when the abundance of organisms is relatively high (at least 2–4%) and the number of possible false positives is marginal. However, as sample complexity increases, raising the number of species and decreasing species abundance, this pipeline tends to produce a higher number of false positives (see Figure 4, Figure 5 and Figure 6). In the second study, we could use both pipelines since we include a set of rRNA sequences. Therefore, the use of a specific rRNA database, such as SILVA or RDP, is recommended. However, we have also used a database of whole genomes, RefSeq, to evaluate the degree of accuracy and the computational performance by Kraken2 with this large database. It is interesting to reflect on GenBank-based databases compared to those based on RefSeq-based information. Databases based on GenBank rely on the correct taxonomic identification and annotation provided by the submitter, so if there is an error, e.g., in labelling, the researcher himself is responsible for correcting it; GenBank can only delete the entry or flag it. In addition, many of these entries are drafts [182]. To avoid such errors, NCBI now performs a series of quality checks when genomes are submitted to ensure that the submitted genomes are not assigned to the wrong species [115]. Another major problem is that the vast majority of GenBank genomes are drafts, in which the chromosomes are fragmented into several contigs and sometimes some of them are contaminants, i.e., they may not belong to the species supposedly sequenced, even though all contigs are assigned to the same species. GenBank itself performs contaminant screening on all assemblies, and contigs that appear to be contaminants are reported to the submitter, who is encouraged to remove and resubmit them. However, despite the efforts of GenBank curators, thousands of contaminants have already crept into the draft genome data causing problems that are difficult to detect when reanalysed [65]. To remedy all these errors, the RefSeq database is attempting to filter GenBank sequences, including viruses, and run them through additional automated filters to produce a more curated genomic resource [187]. Some of the reasons for genome exclusion are overly fragmented assemblies or information derived from a metagenome.

At the genus level, we observe that QIIME2 using SILVA was able to obtain the total of the represented in the sample plus 9% false positives, while Kraken2, together with the same database, was not able to detect one of the genera and included ~7% false positives. Finally, using Kraken with RefSeq as a reference, it was able to detect 51 of the 58 genera, including a false positive rate close to 7%. If instead of taking into account the taxonomic identification, we look at the abundance obtained by these programs with respect to the organisms present (see Figure 9), we observe that the combination of Kraken2 with the SILVA database provided the best results, as well as being the least computationally expensive (see Figure 13).

At the species level (see Figure 10), we found some interesting results given that the Kraken pipeline together with the SILVA database did not provide any results. In the case of QIIME2, using this database, the results showed values very far from the real ones, while Kraken2 with RefSeq was the one that provided the results more in line with reality.

Finally, we processed 16S and RNASeq data from a set of endometrial cancer samples as well as healthy tissue samples. The first thing to note is that, at species level, Kraken2 together with SILVA does not provide any results. To obtain results with this database, we have to process 16S samples with QIIME2.

On the other hand, we did get results from all three types of analysis at the genus level. It is noteworthy that QIIME always provided higher numbers of genera in tumour samples compared to healthy samples, which partly contradicts the assertion of other authors that reduced microbial diversity is often associated with chronic diseases such as cancer [188,189].

Considering the literature, we note that at the genus level, both obtained results in line with other authors, given that the abundant genera in endometrium and/or endometrial cancer are Bacteroides, Prevotella, Pseudomonas, Acinetobacter and Rheinheimera [172,190] (see Figure 11). In terms of species, those provided by QIIME2 do not appear to be related to this tissue, nor to this disease. However, the species provided by Kraken using Refseq as a reference do resemble some known species, such as *E. coli* as well as various Acinetobacters and Pseudomonas [172,190] (see Figure 12).

Consequently, after the results obtained, we can assert that at the genus level, the use of SILVA provides the best results, and if this is combined with Kraken2 we will have a better correlation with reality and with less computational cost and computational time. In the case of species, we conclude that the combination of Kraken2 with a whole genome database such as RefSeq generates the most accurate results.

## 5. Conclusions

The knowledge of microbiomes has been made possible thanks to the research of many groups during recent years, associated in many cases with the creation of consortia. All of this has led to the growth of the available databases and bioinformatics tools to analyse all types of samples in an increasingly accurate way. However, the decisions to be considered in each project are very complex, ranging from how to analyse the data according to specific objectives, computational environment, target taxa and other preferences, to the appropriate experimental design and the tools to be used to obtain reproducible and quality data. Nevertheless, many taxonomic classifiers are still burdened by a high number of false positives, or taxonomic misclassification of microorganisms with low abundance, and identification below the species level must be addressed, in addition to database cleaning and correct taxonomic assignment of the microorganisms that are included. The issues raised throughout this manuscript may be resolved over time, but while the data are in a constant state of flux, users should be aware of these issues so that potential pitfalls can be avoided when analysing large and complex metagenomic datasets.

## Figures and Tables

**Figure 1 genes-13-02280-f001:**
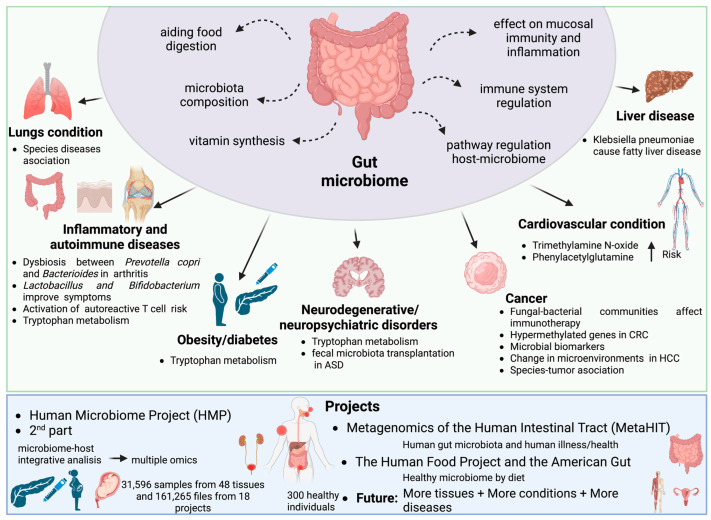
**Effects of the gut microbiome on diseases and international metagenomics and metatranscriptomics consortia and projects**. We collected examples for different diseases regarding a variety of organs and the implications of microorganisms for prognosis and treatment, most of them derived from gut microbiome. For instance, some species has been linked to lung disease [36,37]. Inflammatory and immune diseases have a strong relationship with microorganisms [38,39,40,41,42,43,44], as well as diabetes [44], neurodegenerative and neuropsychiatric disorders [44,45,46], cancer [47,48,49,50], cardiovascular susceptibility [51,52] and liver disease [53]. There are several consortia whose aim is to obtain microorganism information from different types of samples, as de Human microbiome project [54,55,56,57,58,59,60], 2nd part [61,62] the metagenomics of the human intestinal tract [63], the Human Food Project and the American gut [64] and others. CRC: colorectal cancer, HCC: hepatocellular carcinoma, ASD: Autism Spectrum Disorders. Created with BioRender.com accessed on 17 October 2022.

**Figure 2 genes-13-02280-f002:**
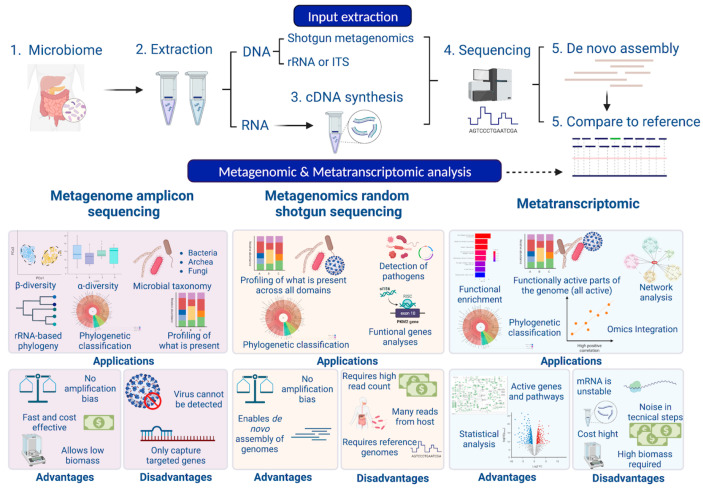
**Workflow of a metagenomic analysis: laboratory and bioinformatic analysis**. 16S rRNA/mRNA/DNA is extracted. Subsequently, they are sequenced. The raw data files are processed using different protocols. On the one hand, de novo assembly can be performed for unknown sequences. On the other hand, samples can be processed directly using different tools. The advantages and disadvantages of the different methodological approaches are presented. Figure created with BioRender.com accessed on 17 October 2022.

**Figure 3 genes-13-02280-f003:**
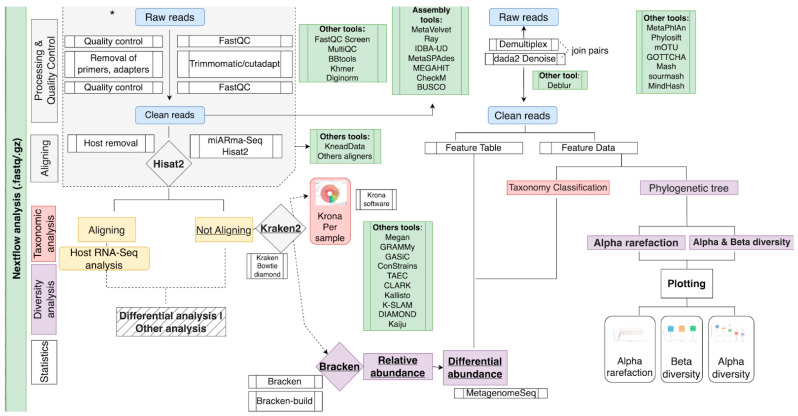
**Overall summary of the Nextflow and QIIME 2 pipeline.** The pipeline works in two stages: (1) Pre-processing stage: importing sequences, quality control, different sequences removal, host sequences removal; (2) Nextflow analysis: QIIME2 or Kraken2 relative abundance analysis and differential abundance and statistics analysis using the metagenomeSeq package in R. The main inputs of the Nextflow pipeline are fastq files from RNASeq pre-processing or 16S fastq files. The box marked with ‘*’ is not included in the Nextflow code. Green boxes show other alternative tools not included in these pipelines.

**Figure 4 genes-13-02280-f004:**
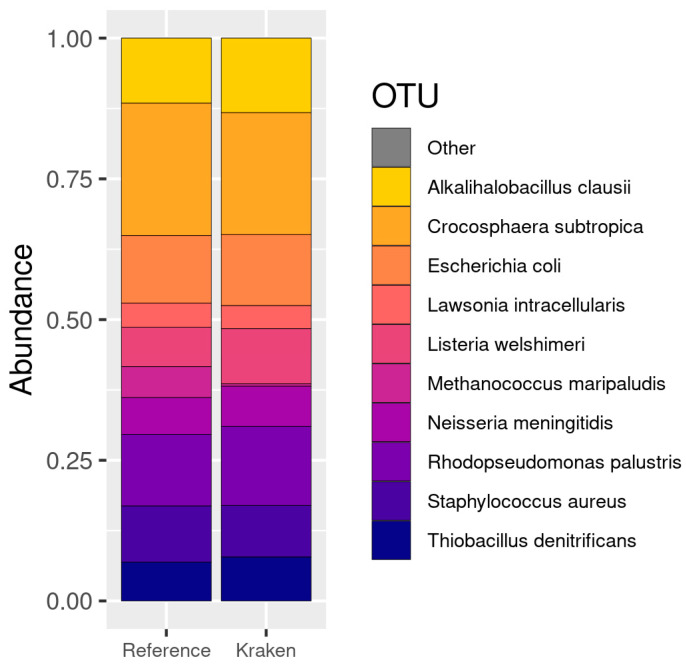
**Stacked barplot depicting relative abundance between Reference abundance and Kraken/Bracken pipeline in the 10 species data from Mendo and collaborators**. Each vertical bar depicts the relative abundance according to Reference data and Kraken/Bracken pipeline analysis (Kraken bar). Counts associated with false positive OTUs are coloured in grey. Species names from the legend along with the associated counts are available in Appendix A.

**Figure 5 genes-13-02280-f005:**
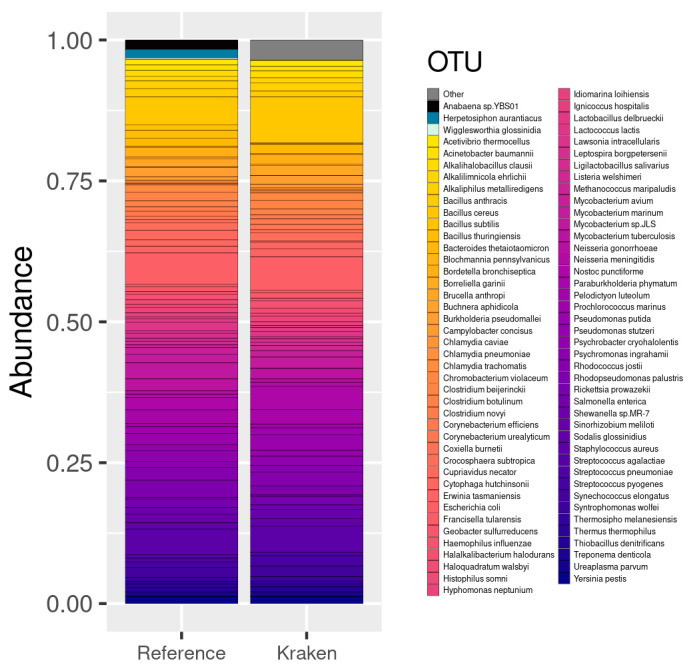
**Stacked barplot depicting relative abundance between Reference abundance and Kraken/Bracken pipeline in the 100 species data from Mendo and collaborators**. Each vertical bar depicts the relative abundance according to Reference data and Kraken/Bracken pipeline analysis (Kraken bar). Counts associated with false positive OTUs are coloured in grey. A different colour palette was used to represent the reference species not detected by the pipeline. The list of OTUs in the legend is included in Appendix A.

**Figure 6 genes-13-02280-f006:**
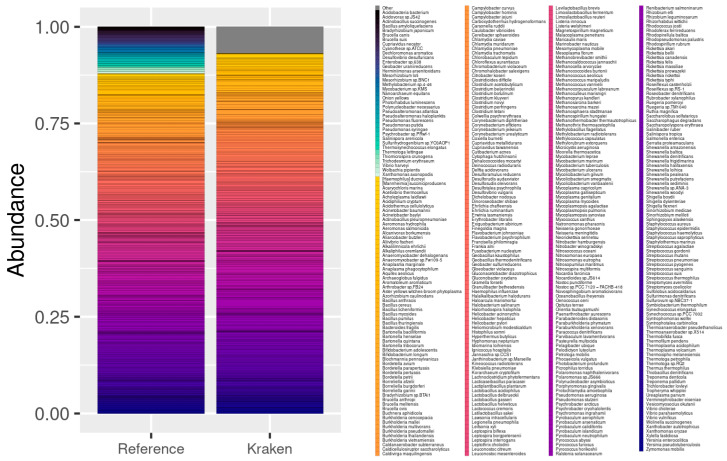
**Stacked barplot depicting relative abundance between Reference abundance and Kraken/Bracken pipeline in the 400 species data from Mendo and collaborators**. Each vertical bar depicts the relative abundance according to Reference data and Kraken/Bracken pipeline analysis (Kraken bar). Counts associated with false positive OTUs are coloured in grey. A different colour palette was used to represent the reference species not detected by the pipeline. The list of OTUs in the legend is included in Appendix A.

**Figure 7 genes-13-02280-f007:**
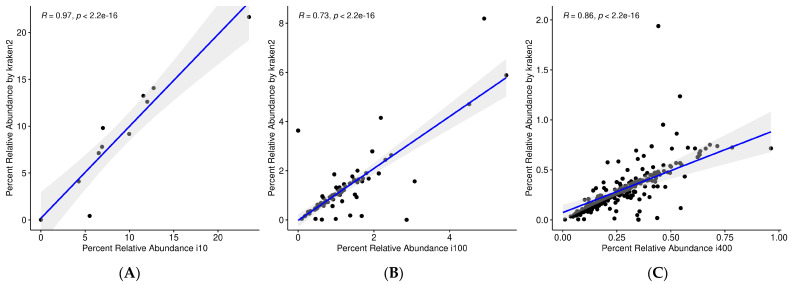
**Correlation plots between the relative abundance of microorganisms detected by Kraken/Bracken pipeline and Reference**. Each plot represents the Spearman correlation between the relative abundance in the 10 species (**A**), 100 species (**B**) and 400 species (**C**) datasets from Mendo and collaborators, along with the obtained correlation coefficient.

**Figure 8 genes-13-02280-f008:**
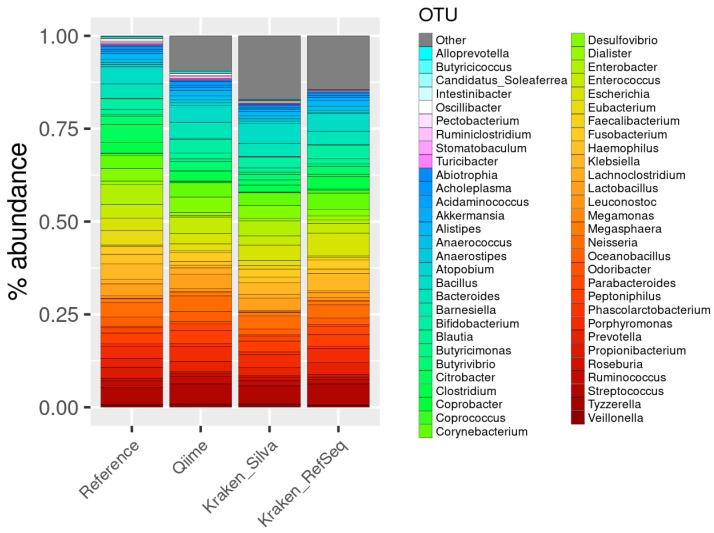
**Stacked barplot depicting relative abundance between Reference abundance, QIIME2, Kraken/Bracken pipeline with SILVA database and Kraken/Bracken pipeline with GenBank repository in gut data from Almeida and collaborators dataset containing 58 genera and 500 species**. Each vertical bar depicts the relative abundance according to Reference data, QIIME2 and Kraken/Bracken pipeline analysis (Kraken bar). Counts associated with false positive OTUs are coloured in grey. A different colour palette was used to represent the reference species not detected by any pipeline. The list of OTUs in the legend is included in Appendix A.

**Figure 9 genes-13-02280-f009:**
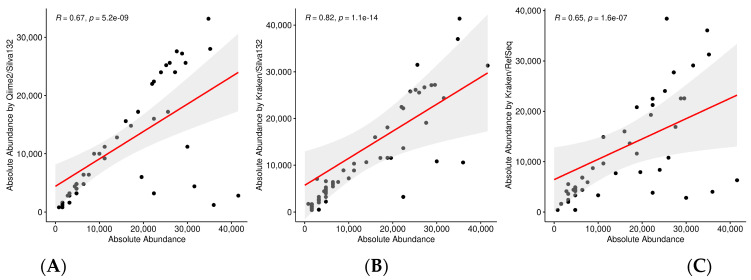
**Correlation plots between the relative abundance of genus detected in Reference and QIIME2 or Kraken/Bracken pipeline with SILVA database and RefSeq repository**. Each plot represents the Spearman correlation between the relative abundance of gut species from Almeida and collaborators detected by QIIME2 (**A**), Kraken/Bracken pipeline with SILVA database (**B**) and Kraken/Bracken pipeline with RefSeq repository (**C**) with its own value of correlation coefficient.

**Figure 10 genes-13-02280-f010:**
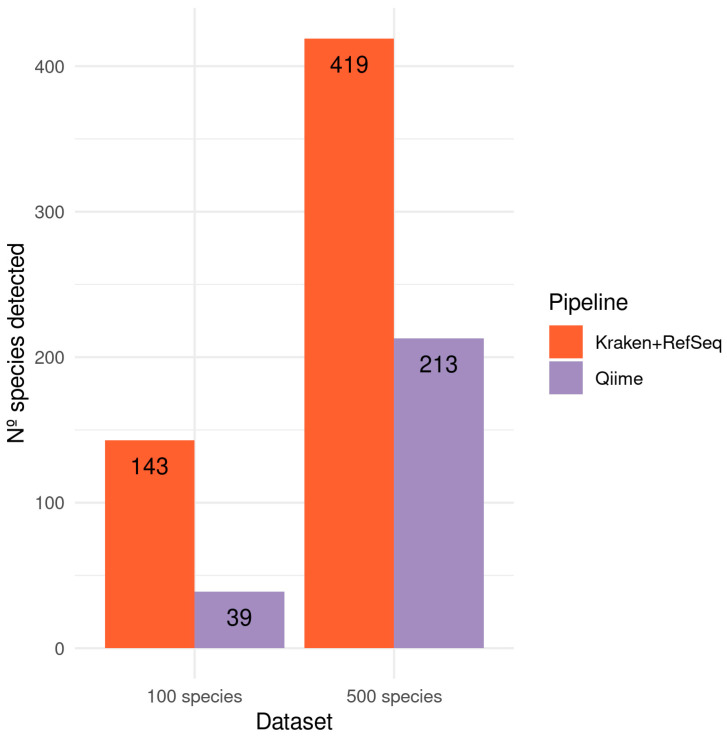
**Number of species identified by Kraken/Bracken pipeline and QIIME2**. Barplot showing the number of microorganisms species detected by Kraken/Bracken pipeline (using RefSeq database) and QIIME2 in gut data from Almeida and collaborators.

**Figure 11 genes-13-02280-f011:**
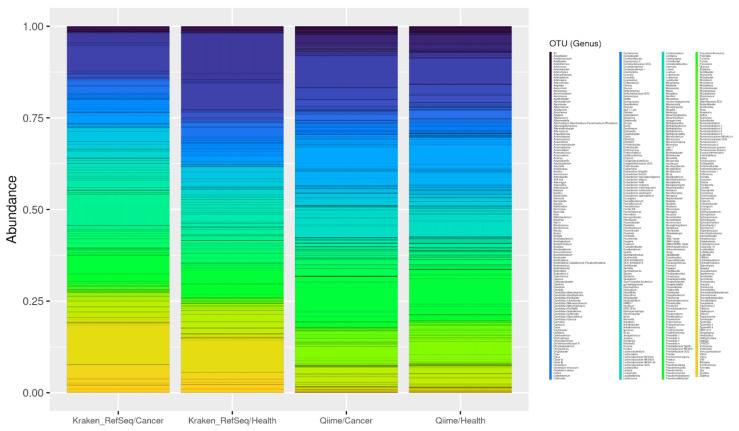
Number of genera identified for tumour/healthy samples by the Kraken/Bracken (in RNA-Seq) and the QIIME2 (in 16S) pipelines. Barplot showing the number of microorganism genera detected by the Kraken/Bracken pipeline (using RefSeq database) and the QIIME2 pipeline (using SILVA) in an endometrial cancer study by Li C. et al. The list of OTUs in the legend is included in Appendix A.

**Figure 12 genes-13-02280-f012:**
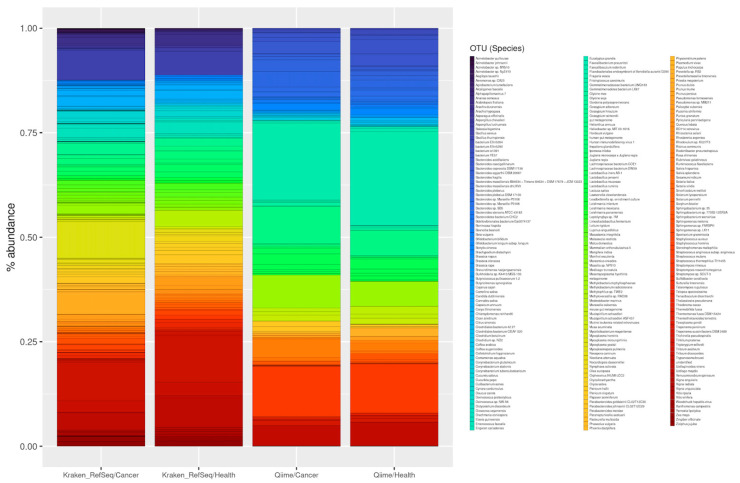
Number of species identified for tumour/healthy samples by the Kraken/Bracken (in RNA-Seq) and the QIIME2 (in 16S) pipelines. Barplot showing the number of microorganism species detected by the Kraken/Bracken pipeline (using RefSeq database) and the QIIME2 pipeline (using SILVA) in an endometrial cancer study by Li C. et al. The list of OTUs in the legend is included in Appendix A.

**Figure 13 genes-13-02280-f013:**
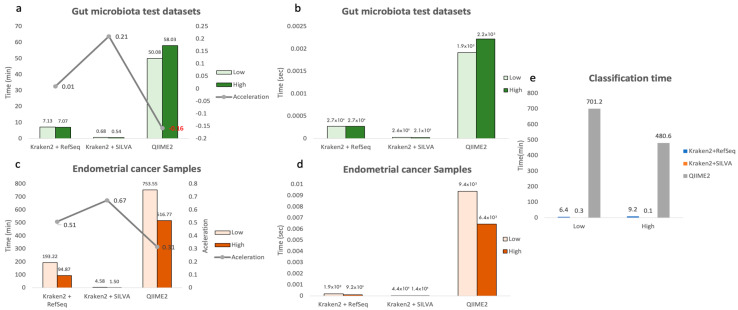
**Computational times**. (**a**,**c**) Total execution time, in minutes, for each pipeline (Kraken, QIIME), database (RefSeq, SILVA132) and computational profile described (low, high), for both the endometrial cancer dataset (**a**) and the gut microbiota test dataset (**c**). These two figures also show, with a grey line, the acceleration values obtained when switching from the low to the high profile. (**b**,**d**) Average processing time per read used by each pipeline. This was calculated by dividing the total time by the number of reads analysed in each run. (**e**) Time used to classify the reads in the database. The times referring to Kraken refer to the average classification time for each sample, as this processing can be parallelised. However, in the case of QIIME, the time spent for all samples is represented, since, as mentioned above, this process is executed in a linear way in Nextflow.

**Table 2 genes-13-02280-t002:** Modifiable parameters of our Kraken/Bracken pipeline.

Parameter	Function	Command
Confidence score threshold	kraken	- confidence
Read length of the input data	bracken-build	- l
kmer length of the reference database	bracken-build	- k
Read length of the input data	bracken	- r
Taxonomic level to filter by	bracken	- l
Threshold for bracken filter	bracken	- t

**Table 3 genes-13-02280-t003:** Modifiable parameters of our QIIME2 pipeline.

Parameter	Function	Command
Format of the input data (casava format, singled/paired, demultiplexed)	import	- input-format
Position to trim reads	dada2 denoise	- p-trim-left
Position to truncate reads	dada2 denoise	- p-trunc-len
Method used to remove chimeras	dada2 denoise	- p-chimera-method
Frequency that each sample should be rarefied	diversity	- p-sampling-depth
Taxonomic level to filter by	collapse	- p-level

## Data Availability

All datasets analysed in this manuscript are available. The first one from Mende et al is located at http://www.bork.embl.de/~mende/simulated_data/. The dataset by Almeida et al. is available at http://ftp.ebi.ac.uk/pub/databases/metagenomics/taxon_benchmarking. The Li C. et al. dataset from cancer was downloaded from the SRA portal at: https://www.ncbi.nlm.nih.gov/bioproject/PRJNA750303. Finally, the code generated in this manuscript is available at the GitHub repository https://github.com/BioinfoIPBLN/16S-Metatranscriptomic-Analysis.

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
