# Peer review of "Comparison of Metagenomics and Metatranscriptomics Tools: A Guide to Making the Right Choice"

_genes, 2022, doi:10.3390/genes13122280_

Round 1

Reviewer 1 Report

The manuscript of Terrón-Camero et al. is dedicated to an up-to-date topic: of metagenomics and metatranscriptomics tools. The authors develop two pipelines to compare different tools. Moreover, several broad aspects of metagenomics and metatranscriptomics are considered. The topic is broad and complicated so tables and figures are very helpful for understanding.

Major comments:

1. The manuscript is quite long. I would shorten it significantly and highlight the original research part. Otherwise, it could be divided into a review and an original article.

I would suggest significantly shortening or even deleting parts 1 and 1.1. Part 1.1 could be reduced to a table.

2. The link to the code given in the abstract is leading to an error https://github.com/BioinfoIPBLN/16S-Metatranscriptomic-Analysis 

Please provide a proper link to the code.

3. In some parts the manuscript is unstructured and returns to the topic that has already been described previously.

4. English editing would improve the quality

Minor comments:

Line 62-as it is presenting the author's original work I would not call it a review

Line 57-“In the last 5 years, more than 4300 articles focusing on the gut microbiota 57 have been published „ how was it checked? PubMed?

Line 79-"The role of the gut microbiome in inflammatory and autoimmune disease models are 79 well-characterized" -I would argue with it. Please provide examples and more than a reference.

Lines 487-500-these tools can be discussed in the Discussion

Section 1.2 Lines 184-194-I would delete this part and just present the tools

Section 1.3 A graph illustrating the workflow and listing tools would be useful

2.1.3. Description of experimental samples – I cannot see a link to these samples (databank)

figure 2 is crucial but is too small to comment on it and see it clearly

Computational times are very important and figures illustrating them should be included in the main text. Now Supplementary figure 1 is as a separate figure and Excel as supplementary Table 7. Please include it only in the Figure

Lines 1046-1049 I think that it is an instruction for the author, it should be deleted.

Discussion: Please discuss the reference database issue here, not in the introduction. Regarding computational timing, are there any examples of parallel programming that succeeded in genetic sequences?

Conclusions are too long and could be reduced to 3-4 most important points

Author Response

The manuscript of Terrón-Camero et al. is dedicated to an up-to-date topic: of metagenomics and metatranscriptomics tools. The authors develop two pipelines to compare different tools. Moreover, several broad aspects of metagenomics and metatranscriptomics are considered. The topic is broad and complicated so tables and figures are very helpful for understanding.

Major comments:

  1. The manuscript is quite long. I would shorten it significantly and highlight the original research part. Otherwise, it could be divided into a review and an original article.

I would suggest significantly shortening or even deleting parts 1 and 1.1. Part 1.1 could be reduced to a table.

Although the journal has no word/page or figure/table limit, such a long review can be difficult to read. When we submitted the original manuscript, we could not find any way to shorten the text. However, thanks to your suggestions we have eliminated some parts of the introduction and others have been added in the Figure 1. This allows us to present the same information but taking less space.

Again, thanks for the suggestion

  1. The link to the code given in the abstract is leading to an error https://github.com/BioinfoIPBLN/16S-Metatranscriptomic-Analysis

Please provide a proper link to the code.

Thank you very much for your comment, we took too long to publish the repository as we include a “README” file to explain all possibilities that both pipelines offer. Besides we have now included in the repository possible dependencies and references.

  1. In some parts the manuscript is unstructured and returns to the topic that has already been described previously.

Thanks again, this comment, together with the first one, has allowed us to reduce the size of the manuscript. We have thoroughly revised the entire manuscript to eliminate possible repetitions. In this way the article reads more fluently.

  1. English editing would improve the quality

Minor comments:

Line 62-as it is presenting the author's original work I would not call it a review

This is a good point. Our group has been working for a long time in the field of metagenomics and metatranscriptomics, so from MDPI we received the proposal to publish a review of the current state of the art, providing a "comparison" between the two strategies and the most used tools in both. Hence the idea of making a benchmark with different datasets (and as bioinformaticians, to provide freely the software developed). We will talk to the Gene Editorial Office, in case it is more convenient (and possible), to publish this manuscript as a scientific article. In the meantime, the phrase: In this review has been changed to In this manuscript

Line 57-“In the last 5 years, more than 4300 articles focusing on the gut microbiota 57 have been published „ how was it checked? PubMed?

Thank you for finding this "error". The number of publications has been obtained by searching in PubMed, so this has been included in the sentence

Line 79-"The role of the gut microbiome in inflammatory and autoimmune disease models are 79 well-characterized" -I would argue with it. Please provide examples and more than a reference.

Thank you for the suggestion, this information has been included in the new Figure 1. Examples and references have been included of how certain specific microorganisms including Lactobacillus casei and Bifidobacteria, are very beneficial in reducing inflammation in autoimmune diseases such as rheumatoid arthritis. However, a dysbiosis associated with an increase in P. copri is associated with a worse prognosis in the development of the disease.

There are more than 3000 publications in PubMed regarding this field of study. In our manuscript, we have highlighted some of them. However, there are many more original articles and reviews of great importance. Some examples are the reviews by Konig in 2020 about the microbiome in autoimmune rheumatic disease, or the one published by Clemente in 2018 about the role of the gut microbiome in systemic inflammatory disease.

Lines 487-500-these tools can be discussed in the Discussion

They have been included in the discussion in lines from 896 to 903.

Section 1.2 Lines 184-194-I would delete this part and just present the tools

            Thanks for the suggestion, we have removed this paragraph from the manuscript

Section 1.3 A graph illustrating the workflow and listing tools would be useful

            In figure 4, which contained the workflow and the software used, we have added the list of the other tools mentioned in the text (and in table 1). In this way it is clear which tools can be used and the characteristics of each one of them.

2.1.3. Description of experimental samples – I cannot see a link to these samples (databank)

Thank you for your comment. We have included the requested information: The samples are from the work of Li Chao et al, and were downloaded from the SRA portal from the following link: https://www.ncbi.nlm.nih.gov/bioproject/PRJNA750303.

This information has been included in the manuscript in lines 435 to 437.

figure 2 is crucial but is too small to comment on it and see it clearly

We are sorry if the image does not display correctly. The figures included in the main text are of small size and low quality so that the document would not be too heavy. However, together with the manuscript we included additional figures of higher quality (300dpi).

We have included in the text the higher quality/sizes figures that allow us to upload the article without problems but I hope that the journal will make available the high-quality figures that are included additionally.

Computational times are very important and figures illustrating them should be included in the main text. Now Supplementary figure 1 is as a separate figure and Excel as supplementary Table 7. Please include it only in the Figure

Thanks for the suggestion. We have made 5 different graphs with the most important aspects of the Supplementary table 7 and it has been added in the main text as Figure 13.

Lines 1046-1049 I think that it is an instruction for the author, it should be deleted.

Thank you for your comment and I hope you accept our apology. The text has been removed and the manuscript has been thoroughly revised so that this will not happen again.

Discussion: Please discuss the reference database issue here, not in the introduction.

The text referring to the database issue now appears in the discussion section between the 912 and 959 and from 968 to 990.

Regarding computational timing, are there any examples of parallel programming that succeeded in genetic sequences?

This is a very good question that can have many different nuances. If we talk about parallelization as the ability to perform a job "in a split and separate way", algorithms like blast years ago included a parameter to use more than one CPU in parallel (threads).

If we understand parallel as the way to automate a computation so that it is divided and executed in parallel, there are fewer examples. The technology we have used in this manuscript, Nextflow, is an example of this parallelization. This program automates parallel computation to increase speed and performance. On the nfs-core site there are several examples of pipelines where genetic sequences are processed in parallel. In addition to nexflow, there are other technologies not as mature, as an example: cromwell: https://github.com/broadinstitute/cromwell

Conclusions are too long and could be reduced to 3-4 most important points

Thank you for this comment which has helped us to reduce the size of the text. The conclusions have been reduced

Reviewer 2 Report

This review article focused on comparison of bioinformatics tools and databases that can be used for 16S, metagenomics and metatranscriptomics data analysis. The main topic of this review to make correct choice of tools accordingly is great, and much information regarding analysis was included. Yet there are many aspects needed to be fixed. Below are my detailed suggestions:

1. Title: The review mainly focused on human microbiome analysis, i.e., not environmental microbiome that is not included in the introduction and does not need analysis steps such as host removal. Please indicate in the title that this review is human microbiome analysis focused.

2. Table 1 is very concerning. The author indicated it was adapted from another published review paper, yet many sentences kept the same. Please revise using your own words/phrases to make this content considered “adapted”.

3. Figure 1 and 2: Please make them bigger. Very hard to see.

4. Line 271- 286: I do not think the description of metaproteomics/metametabolomics fit the topic of this review. Please consider shorten this part and remove them from Table 1.

5. Line 357-375: It looks like the author wrongly saved repeated contents. Please revise.

6. Table 2: Content is missing in “Assembly for Taxonomic classification”.

7. Figure 5: consider remove the legends, instead, put them into supplemental materials.

8. Line 681: Is the Chao Li et al. study correct for this mentioned dataset? In the results part of the EC and HC data analysis the author said data was from Mikamo et al.

9. Line 865: Please clarify in the result section that what the results of each dataset re-analysis mean for readers/users to choose the right tool(s) for their own analysis. Just with these comparison results of how many taxa was obtained from each tool/database did not seem to provide enough information for the readers regarding “a guide to make the right choice”.

10. Line 1046-1049: Did the author check the contents before submitting?

Author Response

This review article focused on comparison of bioinformatics tools and databases that can be used for 16S, metagenomics and metatranscriptomics data analysis. The main topic of this review to make correct choice of tools accordingly is great, and much information regarding analysis was included. Yet there are many aspects needed to be fixed. Below are my detailed suggestions:

  1. Title: The review mainly focused on human microbiome analysis, i.e., not environmental microbiome that is not included in the introduction and does not need analysis steps such as host removal. Please indicate in the title that this review is human microbiome analysis focused.

Thanks for the suggestion. It is true that the introduction primarily mentions diseases in humans (although the majority of this text has now been removed and placed in a figure at the request of reviewer 1).

However only in the case of using RNASeq data to do a transcriptomics study is it necessary to remove the host. Therefore, of the three datasets analyzed only the last one needs this step. In that sense, the tools discussed in the manuscript, as well as the pipelines provided, are valid without any extra modification, for the analysis of environmental or any other kind of samples.

Based on this, we believe that changing the title to include Human would not reflect well the spirit of the manuscript and could prevent possible researchers working on other organisms from reading this article and using the code we have created.

  1. Table 1 is very concerning. The author indicated it was adapted from another published review paper, yet many sentences kept the same. Please revise using your own words/phrases to make this content considered “adapted”.

Thank you very much for your suggestion, we have eliminated it and we have combined the information in the table along with the Figure 2. Your suggestion has helped us to make the ideas more concrete and to express them in the figure in a more visual, original and orderly way.

  1. Figure 1 and 2: Please make them bigger. Very hard to see.

We are sorry if the image does not display correctly. The figures included in the main text are of small size and low quality so that the document would not be too big. However, together with the manuscript we included the figures in higher quality (300dpi).

We have included in the text the higher quality figures that allow us to upload the article without problems but I hope that the journal will make available the high-quality figures that are included additionally.

  1. Line 271- 286: I do not think the description of metaproteomics/metametabolomics fit the topic of this review. Please consider shorten this part and remove them from Table 1.

We agree with your point, and we have reduced the part dedicated to metaproteomics/metametabolomics. The idea that we were most interested in capturing in this section, is the importance of integrating multiple "omic" approaches. However, it is possible that we may have focused too much on two techniques that we do not discuss in the following sections

  1. Line 357-375: It looks like the author wrongly saved repeated contents. Please revise.

Thank you for your comment and I hope you accept our apology. The text has been removed and the manuscript has been thoroughly revised so that this will not happen again.

  1. Table 2: Content is missing in “Assembly for Taxonomic classification”.

Thanks again for this comment, we are sorry for this mistake. We certainly had a lot of problems with the journal template to include tables and figures. That is why tables and figures have been included separately from the text in excel/tiff files.

  1. Figure 5: consider remove the legends, instead, put them into supplemental materials.

Thank you for your comment. The names that appear in the figure legends appear in the supplementary tables associated with each figure. It is true that in the written format (PDF) some names are difficult to read, but in the high-quality images to be included in the final document, they could be read without problems. Additionally, as Genes-MDPI offers the option to read the complete text in HTML format, in that case the figures appear in a larger size and the names can be read easily.

  1. Line 681: Is the Chao Li et al. study correct for this mentioned dataset? In the results part of the EC and HC data analysis the author said data was from Mikamo et al.

Thank you very much for finding that error. The data were obtained from the article by Li Chao. et al. This error has been corrected and the information has been included in the line 429 and 745

  1. Line 865: Please clarify in the result section that what the results of each dataset re-analysis mean for readers/users to choose the right tool(s) for their own analysis. Just with these comparison results of how many taxa was obtained from each tool/database did not seem to provide enough information for the readers regarding “a guide to make the right choice”.

Thanks for the suggestion, we think it is very useful and certainly helps to make the right choice. In the part of the results a text has been included as a summary to show which of the tools has provided the best results. In the case of the data by Mende et al. we have included a summary from lines 676 to 683. In the case of Almeida et al it appears in lines 732 and 743 and in the dataset of Li C. et al in lines 804 to 812.

  1. Line 1046-1049: Did the author check the contents before submitting?

We are sorry again for this error with the template. We have corrected that error and others we have found. Also, the manuscript has been thoroughly revised so that this will not happen again.

Round 2

Reviewer 1 Report

I am satisfied with the revision and have no further comments.

Reviewer 2 Report

I have no further comment on this revision.